# Remote Actuation of Apoptosis in Liver Cancer Cells via Magneto-Mechanical Modulation of Iron Oxide Nanoparticles

**DOI:** 10.3390/cancers11121873

**Published:** 2019-11-26

**Authors:** Oleg Lunov, Mariia Uzhytchak, Barbora Smolková, Mariia Lunova, Milan Jirsa, Nora M. Dempsey, André L. Dias, Marlio Bonfim, Martin Hof, Piotr Jurkiewicz, Yuri Petrenko, Šárka Kubinová, Alexandr Dejneka

**Affiliations:** 1Institute of Physics of the Czech Academy of Sciences, 18221 Prague, Czech Republic; uzhytchak@fzu.cz (M.U.); smolkova@fzu.cz (B.S.); mariialunova@googlemail.com (M.L.); sarka.kubinova@iem.cas.cz (Š.K.); dejneka@fzu.cz (A.D.); 2Institute for Clinical & Experimental Medicine (IKEM), 14021 Prague, Czech Republic; miji@ikem.cz; 3Institut Néel, Grenoble INP, CNRS, Université Grenoble Alpes, 38000 Grenoble, France; nora.dempsey@neel.cnrs.fr (N.M.D.); andre.dias@neel.cnrs.fr (A.L.D.); 4Universidade Federal do Paraná, DELT, Curitiba 81531-980, Brazil; marliob@eletrica.ufpr.br; 5J. Heyrovský Institute of Physical Chemistry of the Czech Academy of Sciences, 18223 Prague, Czech Republic; martin.hof@jh-inst.cas.cz (M.H.); piotr.jurkiewicz@jh-inst.cas.cz (P.J.); 6Institute of Experimental Medicine of the Czech Academy of Sciences, 14220 Prague, Czech Republic; yuriy.petrenko@iem.cas.cz

**Keywords:** pulsed magnetic field, lysosomal membrane permeabilization, magnetic nanoparticles, lysosomal death pathways, apoptosis

## Abstract

Lysosome-activated apoptosis represents an alternative method of overcoming tumor resistance compared to traditional forms of treatment. Pulsed magnetic fields open a new avenue for controlled and targeted initiation of lysosomal permeabilization in cancer cells via mechanical actuation of magnetic nanomaterials. In this study we used a noninvasive tool; namely, a benchtop pulsed magnetic system, which enabled remote activation of apoptosis in liver cancer cells. The magnetic system we designed represents a platform that can be used in a wide range of biomedical applications. We show that liver cancer cells can be loaded with superparamagnetic iron oxide nanoparticles (SPIONs). SPIONs retained in lysosomal compartments can be effectively actuated with a high intensity (up to 8 T), short pulse width (~15 µs), pulsed magnetic field (PMF), resulting in lysosomal membrane permeabilization (LMP) in cancer cells. We revealed that SPION-loaded lysosomes undergo LMP by assessing an increase in the cytosolic activity of the lysosomal cathepsin B. The extent of cell death induced by LMP correlated with the accumulation of reactive oxygen species in cells. LMP was achieved for estimated forces of 700 pN and higher. Furthermore, we validated our approach on a three-dimensional cellular culture model to be able to mimic in vivo conditions. Overall, our results show that PMF treatment of SPION-loaded lysosomes can be utilized as a noninvasive tool to remotely induce apoptosis.

## 1. Introduction

Iron oxide magnetic nanoparticles have been extensively utilized in a wide range of biomedical applications [1,2]. Unique physicochemical properties of superparamagnetic iron oxide nanoparticles (SPIONs) enable their use in diverse biomedical fields. SPIONs have been found to be very useful as nanosensors in in vitro diagnostic tests [3,4], contrast agents in in vivo imaging [5,6], therapeutic modalities in magnetic fluid hyperthermia [7], and drug carriers in targeted drug delivery [8]. Recent studies proposed the external application of magnetic fields as a simple, noninvasive tool to remotely control cell behavior via SPION mediated magneto-mechanical actuation [2,9]. Alternating magnetic fields have been successfully utilized for magneto-mechanical remote-control of SPIONs in a wide range of biomedical applications; e.g., magnetic tweezers, nanosensing, magnetic cell separation, specific delivery of genes and therapeutic agents, and mechanical modulation in cells [2,9,10,11]. Interestingly, magnetic field-induced mechanical destruction of cells or cellular organelles was proposed as a promising potential tool for anti-cancer therapy [12,13,14,15].

It becomes evident that mechanical forces transmitted by magnetic nanoparticles under alternating or dynamic magnetic field treatment are large enough to disrupt lysosomal membranes, resulting in lysosomal membrane permeabilization (LMP) and the subsequent initiation of apoptotic cell death [13,15,16,17,18]. Such treatment requires loading of cells with appropriate magnetic nanoparticles and subsequent application of external alternating or dynamic magnetic fields [13,15,16,17,18]. However, all those studies utilized relatively bulky generators of either an alternating magnetic field with up to ~kW power, a frequency range from a few Hz to hundreds of kHz, and amplitude of magnetic field from 10 to 100 mT; or dynamic fields with low magnetic field strength of about 30 mT [13,15,16,17,18]. The bulky construction of such generators hampers the development of fast benchtop treatment protocols. Additionally, such a treatment of cells requires, typically, 30 min to 1 h of magnetic field exposure [13,15,16,17,18]. Some studies have utilized pulsed magnetic fields (PMFs) produced by large devices composed of moving arrays of bulk permanent magnets or low intensity electro-magnets (30–300 mT) to trigger apoptosis [16,17,19]. In this study we propose a fundamentally different approach of utilizing high intensity (up to 8 T), short pulse width (~15 µs) PMFs produced by a compact, coolant-free system to induce apoptosis in a controlled manner via LMP by transmitting mechanical force to clusters of SPIONs retained in lysosomes.

Although magneto-actuated cell apoptosis was proposed in a number of studies [12,13,15,16,18,20,21], there are only a few reports that estimate the force exerted on magnetic nanoparticles by the applied magnetic fields [9,12]. Additionally, there is a lack of studies attempting to translate magneto-actuated cell apoptosis to in vivo relevant models [12,13,15,16,18,20,21]. Here, we revealed a new mode of magneto-actuated cell apoptosis via application of high intensity PMFs and validated this new approach on three-dimensional cell culture models mimicking in vivo conditions. To the best of our knowledge, this is the first experimental report demonstrating a clear effector mechanism for pulsed magnetic fields with high field gradients.

## 2. Results

### 2.1. Characterization of the Pulsed Magnetic Field (PMF) System and Internalization of SPIONs

We utilized a previously designed and validated PMF benchtop system capable of generating magnetic fields as high as 10 T [22]. A scheme of the PMF benchtop system is presented in Appendix A. In such a system the intensity of the magnetic field pulse generated by the coil is directly proportional to the capacitor’s charge voltage. The experimental measurement and simulation of the drop-off in field intensity with distance from the surface of the coil showed very good agreement [22]. We have already demonstrated the applicability of this PMF system for actuation of SPIONs and SPION clusters via a magnetic gradient force [22]. Positive results from this study prompted us to explore the potential use of this system for remote actuation of cancer cell death via magneto-mechanical modulation of SPIONs. For the experiments reported here, the voltage was set at 550 or 850 V to produce a field of 5.5 or 8.5 T, respectively, at the surface of the coil. The drop off in magnetic field gradient along the coil axis is shown in Appendix A. The rapid drop off in both field and field gradient with distance from the coil means that cells must be positioned close to the coil to benefit from the intense magnetic field and field gradient produced. The field intensity at the position of the cells was estimated to be 5 T and 8 T, for voltages of 550 or 850 V, respectively. For more detailed information about designed PMF benchtop system, please, visit www.pumag.fr [22].

Figure 1a depicts a simulation of the spatial distribution of the norm of the field gradient produced by the PMF system. The force exerted by the system on a target object is proportional to the field gradient [22]. A schematic of the experimental setup used in this study is shown on Figure 1b.

It has been shown that SPIONs can be used to remotely activate apoptosis via LMP [13,15,16,17,18]. For cancer cell labelling, we selected previously characterized carboxydextran-coated SPIONs (mean hydrodynamic diameter of about 60 nm) which we used for biocompatibility screening [23,24,25]. Briefly, the physicochemical characteristics of the SPIONs are summarized in Appendix A. A detailed, full characterization of the SPIONs was reported elsewhere [23,24,26,27,28,29,30]. We previously described endocytosis and cell labelling with these particles [25,26]. Additionally, we showed the feasibility of using static and PMFs to enhance endocytosis of such nanoparticles by different cell types [22,27]. Overall, the selected SPIONs represent well-characterized magnetic nanoparticles that show a robust response to magnetic fields.

Remote actuation of magnetic nanoparticles by external magnetic fields for selective cancer cell treatment was previously utilized on different cancer cell lineages [12,13,14,15,16,17,18]. However, the number of studies comparing nanoparticle-induced LMP on the same cancer model using different cell lines is rather limited. Furthermore, studies utilizing liver cancer cell lines are very limited in number. Thus, in the present study we selected as cell models, two hepatocellular carcinoma (Huh7 and Alexander cells) and one hepatoblastoma (HepG2) cell line.

First of all, we evaluated whether SPIONs could be effectively engulfed by the three liver cancer cell lines. Figure 1c shows representative confocal microscopy images of the distribution of SPIONs (red) inside cells after 1.5 h of incubation. 3D reconstruction of cells, incubated with SPIONs (red) and counterstained with membrane label (green), shows clear intracellular localization of SPIONs (Figure 1c). The punctate lysosomal staining pattern (green) was very similar to labeled SPIONs (red), suggesting that SPIONs are concentrated inside the lysosomes (Figure 1d). Quantitative colocalization analysis confirmed lysosomal localization of SPIONs 1.5 h post incubation in all three cell lines (Figure 1e). The confocal images in Figure 1f illustrate that SPION lysosomal localization is accompanied by an increase of lysosomal size. The increase of lysosomal size was SPION dose-dependent in all three cell lines (Figure 2a).

### 2.2. PMF Stimulation of SPION-Loaded Cells Results in Cell Death

Further, we conducted a proof-of-concept experiment using the Huh7 cell line. We checked cytotoxic effects exerted by PMF. Indeed, cell treatment solely with PMF of 5 T amplitude did not induce noticeable cytotoxic effects on the cells (Figure 2b). This is in line with our previous findings showing no effects of PMF on cellular viability [22]. Furthermore, SPIONs alone did not induce cytotoxicity in Huh7 cells (Figure 2b). On the contrary, Huh7, pre-incubated with 100 μg Fe mL^−1^ SPIONs and then subjected to 5 T PMF, showed significant reduction in viability (Figure 2b). Huh7, pre-incubated with either 10 or 50 μg Fe mL^−1^ SPIONs and then subjected to 5 T PMF, did not show signs of cytotoxicity (Figure 2b). Further we increased the amplitude of PMF up to 8 T, but decreased pre-incubation concentration of SPIONs (50 μg Fe mL^−1^). Indeed, we found that Huh7, pre-incubated with 50 μg Fe mL^−1^ SPIONs and then subjected to 8 T PMF, exhibited significant toxicity (Figure 2c). It is important to check the physiologically relevant concentration of nanoparticles. Persistent nanomaterial delivery problems [31] and rapid clearance from the bloodstream by cells of the mononuclear phagocyte system [32] created a situation where many studies, until now, have been done at much higher doses than are realistic [33,34]. Indeed, after intravenous injection, the blood level of SPIONs may reach 50 μg Fe mL^−1^ and liver accumulation could be as high as 100 μg Fe mL^−1^ [23,32,35,36]. Our findings indicate that loading cells with physiologically relevant concentrations of SPIONs and then subjecting loaded cells to PMF triggers cell death.

One can easily estimate the magnetic gradient force exerted on a SPION (or cluster of SPIONs) by the applied magnetic field using the following equation *F = p_m_∙dB/dz*, where the magnetic moment of the SPION (or cluster of SPIONs) is designated as *p_m_* and *dB/dz* is the magnetic field gradient generated by the coil at the position of the cells. SPIONs exposed to a field of 5–8 T will be saturated. Thus, one can estimate their magnetic moment as *p_m_ = M_s_V*, where *V* is the SPION (or cluster of SPIONs) volume and *M_s_* represents the saturation magnetization of Fe_3_O_4_ (*M_s_* = 80 emu/g = 412 kA/m) [37]. We estimated that the maximum magnetic field gradient (*dB/dz*) generated by the coil at the position of the cells reaches values of ~3400, 2000, and 1000 T/m, for field pulses of 8, 5, and 3 T respectively. Appendix A depicts the drop-off in field gradient with distance from the surface of the coil for a charge of 100 V. Importantly, at the distance of ~1 mm, where cells are located, the gradient is reduced by approximately by 40%. Given that the field intensity and resultant gradient are proportional to the voltage, one can roughly estimate the field gradient at the position of the cells to be ~3400, 2000, and 1000 T/m, for voltages of 850, 550, and 300 V, respectively. Note that even higher field gradients can be achieved with low and/or moderate magnetic field intensities if the size of the field source is significantly reduced [38], but the range of interaction would also be reduced. One can tentatively estimate the force exerted on a cluster of SPIONs as a function of its size, which is taken to be equivalent to that of the lysosome (Figure 2d). As we noted above, accumulation of SPIONs induces an increase in the size of lysosomes (Figure 2a). Specifically, incubation with 100 μg Fe mL^−1^ SPIONs leads to an increase in size of the lysosomes up to 1.2 μm (corresponding to a cluster radius of 600 nm) and incubation with 50 μg Fe mL^−1^ SPIONs—up to 1 μm (corresponding to a cluster radius of 500 nm).

It is worth noting here that a number of studies used either high (>kHz) or low (1–50 Hz) frequency alternating magnetic fields to disrupt membranes of cancer cells via the exertion of mechanical forces on magnetic nanoparticles to induce apoptosis [12,13,15,16,18,20,21]. However, only a few attempted to estimate the force exerted on magnetic nanoparticles by the applied magnetic fields [9,12]. A recent study estimated a force of 500 pN to be the minimal effective force required to induce leakage of lysosomal contents into the cytoplasm [39]. Additionally, one can compare this force with the Stokes’ drag force, *Fs = 6πµRv*, where *Fs* is known as Stokes’ drag force, *µ* represents the viscosity, *R* is the radius of the vesicle, and *v* is the flow velocity. Assuming the lysosome is spherical with a radius of 500 nm, and that it is moving through the cytosol of density 1.3 × 10^−3^ Pa∙s [40] at a velocity of 10 µm/s, one can roughly estimate a drag force of ~0.1 pN. It is obvious that the force exerted on magnetic nanoparticles by the applied magnetic field (500 pN) outweighs the Stokes’ drag force. However, one needs to take into account that lysosomes are not freely moving in the cytosol [41]. Indeed, lysosomes are linked to the cytoskeletal elements [41,42]. Therefore, it is important to assess the hoop stress (*σ*) on a thin-walled lysosomal vesicle. One can roughly derive the hoop stress as *σ = P∙r*/2*∙t*, where *P* is the internal pressure (knowing the exerted force ~500 pN and lysosomal diameter ~1 μm, one can estimate the pressure to be ~600 Pa), *r* represents the radius of the lysosome, *t* is the lysosomal membrane thickness. Assuming the lysosomal radius ~500 nm (Figure 2a) and membrane thickness of ~10 nm [43], we roughly estimated the hoop stress to be about 15 kPa. This estimation is perfectly in line with the recently measured 12 kPa critical pressure required for membrane rupture in endothelial cells [44].

Interestingly, for the SPION cluster radius of 600 nm (corresponding to incubation with 100 μg Fe mL^−1^ SPIONs) and *dB/dz* = 3400 T/m, we estimated the force to be *F* ~1300 pN (Figure 2d). For the SPION cluster radius of 500 nm (corresponding to incubation with 50 μg Fe mL^−1^ SPIONs) and *dB/dz* = 3400 T/m, the force will be *F* ~700 pN (Figure 2d). These values are above the reported minimal effective force required to induce lysosomal disruption [39]. Thus, one can suggest, that in order to induce lysosomal disruption and subsequent cell death, it is necessary to generate a large enough mechanical force enabling loss of membrane integrity. To verify this hypothesis, we pre-incubated the three cell lines with different SPION concentrations and then subjected loaded cells to 8 T PMF. Incubation of all three cell lines with 10, 50, and 100 μg Fe mL^−1^ SPIONs resulted in lysosomal nanoparticle accumulation and cluster radii of 200, 500, and 600 nm (Figure 2a). Therefore, we expected the corresponding force exerted on the SPION clusters to be of the order of 50, 700, and 1300 pN, respectively (Figure 2d). Indeed, we found that all cell lines, pre-incubated with either 50 or 100 μg Fe mL^−1^ SPIONs and then subjected to 8 T PMF, showed significant reduction in viability (Figure 2e). On the contrary, cells pre-incubated with 10 μg Fe mL^−1^ SPIONs and then subjected to 8 T PMF, did not show signs of cytotoxicity (Figure 2e). This finding is perfectly in line with the aforementioned force estimations (Figure 2d). Moreover, if we correlate corresponding loading concentrations of SPIONs to the resultant force exerted on a SPION cluster, we get the direct dependence of viability from the force exerted on a SPION cluster (Appendix A).

### 2.3. Triggering Apoptosis by PMF Stimulation via Disruption of Lysosomes

Our hypothesis was that upon PMF treatment, SPION-loaded lysosomes would be subjected to mechanical stress that in turn would lead to lysosomal leakage and/or dysfunction. Therefore, to investigate such lysosomal damage experimentally, we utilized a well-established methodology based on lysosomotropic dye (AO) labeling [45,46,47]. The principle of AO staining is based on its accumulation in lysosomes, which is associated with the formation of aggregates of AO exhibiting red fluorescence. Upon lysosomal loss of membrane integrity, dye leaks from this compartment into the cytosol, leading to a decrease in red fluorescence. As shown in Figure 3a, cells loaded with 100 μg Fe mL^−1^ SPIONs and exposed to 5 T PMF show a significant decrease in AO red fluorescence, as compared to a control.

It becomes evident that different cell death pathways ranging from classical apoptosis to necrosis are regulated by LMP [48,49,50,51]. Additionally, the mitochondrial membrane permeabilization (MMP) and subsequent caspase-dependent and/or caspase-independent apoptosis pathways are modulated by the degree of LMP [48,49,50,51]. Therefore, the next logical step was to study whether our PMF-induced cell death is associated with the impairment of mitochondrial function. To answer this question, we utilized a well-established methodology based on JC-1 staining [52,53,54]. JC-1 mitochondria membrane potential (ΔmΦ) assessment is possible due to the selectivity of the dye in mitochondria labeling. In mitochondria of healthy cells, JC-1 accumulates and exhibits color change from green to red as ΔmΦ increases. On the contrary, in unhealthy cells with damaged mitochondria and subsequently low ΔmΦ, the dye leaks into the cytosol, leading to a dramatic decrease in red fluorescence. Indeed, cells loaded with 100 μg Fe mL^−1^ SPIONs and exposed to 5 T PMF showed a significant decrease in JC-1 red fluorescence, compared to a control (Figure 3b). These data indicate a reduction of mitochondrial membrane potential and subsequent mitochondrial dysfunction upon PMF treatment of SPION-loaded cells. Importantly, loss of ΔmΦ was detected after initiation of LMP (Figure 3a,b). This finding indicates that initiation of cell death begins in the lysosomes. Further, cells loaded with 100 μg Fe mL^−1^ SPIONs and exposed to 5 T PMF showed early signs of apoptosis (Figure 3c,d). Annexin V/propidium iodide labeling revealed significant phosphatidylserine exposure without a concomitant increase in membrane permeability of the treated cells (Figure 3c,d).

However, early exposure of phosphatidylserine can be associated with necrotic pathways [55,56,57]. Therefore, we assessed effector caspase-3 activity to further validate apoptotic cell death. Caspase-3 activity is driven by apoptogenic factors, such as cytochrome c leakage from the damaged mitochondria [56,58]. Fluorometric analysis of caspase 3 activation in SPION-loaded cells treated with 5 T PMF confirmed that, indeed, PMF treatment led to apoptotic cell death (Figure 3e). In order to support the hypothesis that cell death is dependent on the force exerted on a cluster of SPIONs by the applied magnetic field, we loaded cells with 50 μg Fe mL^−1^ SPIONs and then subjected them to 8 T PMF (Appendix A). Indeed, cells loaded with 50 μg Fe mL^−1^ SPIONs and then subjected to 8 T PMF showed LMP and loss of ΔmΦ (Appendix A). Additionally, 8 T PMF triggered apoptosis in SPION-loaded cells (Appendix A).

Further, to validate that apoptotic activation under PMF treatment of SPION-loaded cells is mediated by LMP, we assessed a more specific marker of lysosomal damage—cathepsin B. Lysosomes retain various enzymes, including proteolytic ones, such as cathepsins. Leakage of cathepsins into the cytosol upon disruption of the lysosomal membrane serves as a specific LMP hallmark [48,49,50]. In order to validate that PMF treatment triggers lysosomal destabilization and/or rupture, we assessed the lysosomal release of cathepsin B by immunofluorescent labeling. In line with the AO destabilization assay (Figure 3a), release of mature lysosomal cathepsin B into the cytosol of the SPION-loaded cells subjected to 8 T PMF became detectable as well (Figure 4a,b). Additionally, the lysosomal release of cathepsin B concurred with the observation of large swollen lysosomes (Figure 4a,c). Indeed, AO destabilization assay (Figure 3a) together with the leakage of cathepsin B into the cytosol (Figure 4a,c) confirmed that PMF treatment results in significant lysosomal destabilization of SPION-loaded cells.

Reactive oxygen species (ROS) have been repeatedly identified as major mediators of the LMP-dependent apoptosis [48,49,50,51]. Thus, our next step was to evaluate the degree of intracellular ROS accumulation upon PMF treatment of SPION-loaded cells. We utilized a previously well-established methodology based on two distinct fluorescent probes [59,60,61,62]. One probe labels total ROS, while the other one is highly-specific against superoxide anion (O_2_^−^). Such a combination allowed us to simultaneously detect changes in the total ROS level as well as superoxide anion. It is worth mentioning that the level of superoxide is an indicator of mitochondrial membrane permeabilization and dysfunction. Indeed, PMF treatment of SPION-loaded cells triggered a dose-dependent accumulation of both total ROS and superoxide (Figure 4d).

Altogether these results demonstrate that PMF-exerted forces on clusters of SPIONs are capable of disrupting lysosomal membranes. Initiated by PMF, lysosomal leakage results in mitochondria damage accompanied with loss of ΔmΦ and excessive ROS accumulation. Mitochondrial dysfunction together with high ROS levels triggered apoptosis (Figure 4e).

### 2.4. PMF Induces Apoptosis in 3D Multicellular Aggregates

Although a number of studies have utilized alternating magnetic fields to disrupt membranes of nanoparticle-loaded cancer cells, there is a lack of studies which attempted to translate such an approach to in vivo relevant models [12,13,15,16,18,20,21]. Indeed, all previous studies were done using standard two-dimensional (2D) cell cultures in vitro. Recent studies indicate that conventional 2D culturing possesses many limitations, which in turn preclude transition of the obtained in vitro results to in vivo relevant models [63,64]. Three-dimensional culture (3D) models provide conditions which are more similar to in vivo systems [63,64]. Thus, to validate our PMF approach to trigger apoptosis in cancer cells, we utilized 3D multicellular aggregates (Figure 5a). Firstly, we checked whether we could efficiently load 3D multicellular aggregates generated from the three cell lines with SPIONs. Indeed, 3D multicellular aggregates ingested significant numbers of SPIONs after 2 h of incubation (Figure 5b). Secondly, uptake of SPIONs was accompanied by lysosomal localization of ingested nanoparticles (Figure 5c). Finally, 3D multicellular aggregates of all three lines loaded with SPIONs and exposed to PMF showed signs of apoptosis (Figure 5d–f and Appendix A).

These results suggest that the force produced by the pulsed field dramatically affects the stability of SPION-loaded lysosomes, prompts apoptotic cell death, and affects the growth of the cancer cell population even in a 3D culture model.

## 3. Discussion

Lysosomal membrane permeabilization in nanoparticle-loaded cells triggered by external magnetic fields has proven its applicability in various cancer models [9,12,13,14,15,16,17,18]. Furthermore, the general concept, that magnetic nanoparticles transmit a large enough mechanical force leading to lysosomal membrane destabilization in the presence of magnetic fields, has been extensively explored and validated [13,15,16,17,18]. However so far, only relatively bulky systems were used in order to generate alternating or dynamic magnetic fields to trigger apoptotic responses in cancer cells [13,15,16,17,18]. Additionally, treatments utilizing alternating or dynamic magnetic fields lasted up to 1 h [13,15,16,17,18]. Therefore, we hypothesized that the application of high intensity pulsed magnetic fields would dramatically reduce the treatment time.

Here, we proposed the use of a benchtop pulsed magnetic field system [22] capable of producing high intensity (8 T), short pulse width (~15 µs) PMFs (Figure 1a and Appendix A). Using this system, we were able to induce a significant degree of cell death in three distinct liver cancer cell lines (Figure 2e). Of note, the exposure to PMF lasted not more than 5 min. Importantly, the degree of cell death was proportional to the concentration of SPIONs used to load cells (Figure 2b,c,e). The degree of cytotoxicity also showed a clear dose dependency, depending on the intensity of PMF treatment (Figure 2b,c and Appendix A).

The degree of cytotoxicity also showed a clear dose dependency, depending on the intensity of PMF treatment (Figure 2b,c and Appendix A). Previously published studies [9,12,13,14,15,16,17,18] and our data taken together clearly indicate that magnetic nanoparticles transmit mechanical forces under external magnetic field treatment, that may lead to subcellular organelle damage and subsequent cell death. However, this force needs to be large enough to be able to induce destabilization and/or damage of cellular membranes. Only a limited number of studies estimated the mechanical forces transmitted by magnetic nanoparticles [9,12,17]. Furthermore, there are only isolated reports estimating the value of mechanical forces needed to initiate lysosomal membrane permeabilization [9,39]. It is worth noting that in silico simulations postulate that relatively small tension and/or shear may induce cell membrane destabilization [65]. In our study we estimated the minimal effective force needed to trigger LMP (Figure 2d and Appendix A). We established that accumulation of SPIONs results in a proportional increase in lysosome size (Figure 2a). The PMF system allows the generation of high intensity short magnetic field pulses in a tunable manner. Thus, by loading cells with different concentrations of SPIONs and applying different pulses, one can easily modulate the resultant mechanical force exerted on lysosomal membranes (Figure 2d and Appendix A). This allowed us to estimate that a force of ~500 pN is required for the initiation of LMP. Interestingly, this force value was perfectly in line with the force calculated to trigger lysosomal permeabilization by carbon nanotubes [39].

It is worth mentioning here that for potential in vivo translation of such cancer treatment, one needs to consider targeting strategies and the selectivity of the treatment. Targeting is crucial in light of recent criticism of different nanoparticle targeting approaches [31]. However, enhanced permeability and retention (EPR) may help to overcome targeting challenges [66]. The EPR effect is based on the preferential passive accumulation of nanoparticles in cancerous tissues due to the enhanced permeability of the vasculature that supplies such tissues [66,67,68]. In fact, liposomal doxorubicin (Doxil™/Caelyx™), that utilizes the EPR effect, was approved by the FDA and is used in clinics as a treatment modality [66,69]. The EPR effect may endow a significant advantage for liver cancer targeting, since dextran-decorated SPIONs are known to be passively accumulated in liver; for a review see [70]. Additionally, cancer treatment strategies involving LMP have selectivity advantages. Recent studies show that lysosomal membranes of cancer cells are mechanically weaker than those of noncancerous cells [71,72]. Thus, cancer cells are more susceptible to LMP. In fact, a number of studies have already utilized this approach to selectively sensitize different cancer cell types to cell death [47,73,74,75,76]. Indeed, several agents that target LMP in cancer are being extensively investigated in preclinical studies [71].

In fact, previous studies did not compare the possibility of inducing lysosomal membrane disruption under magnetic field exposure in distinct cell lines of the same cancer model [13,15,16,17,18]. We validated our approach on three distinct liver cancer (Huh7, HepG2, and Alexander cells) cell lines (Figure 2e). Of note, HepG2 cells are known to bear higher levels of Bcl-2 protein compared with Alexander and Huh7 cells [53,62,77,78]. In fact, Bcl-2 blocks different cell death pathways [79,80,81,82] and stabilizes lysosomes, hampering LMP [83,84]. Thus, high expression levels of Bcl-2 make liver cancer cells resistant to conventional treatments [77,81,82]. However, using our PMF system we could induce cell death even in LMP-resistant HepG2 cells (Figure 2e).

So far, studies on magneto-actuated cell apoptosis are mostly limited to in vitro models without a translation of this therapeutic strategy to in vivo relevant models [12,13,15,16,18,20,21]. Here we showed the feasibility of PMF-induced apoptosis in 3D multicellular aggregates (Figure 5). Of course 3D organoids or other 3D cell culture models do not fully recapitulate real in vivo systems. However, to the best of our knowledge, our study represents a first attempt to translate PMF-induced apoptosis treatment into more complicated culturing systems then conventional 2D cell cultures.

## 4. Materials and Methods

### 4.1. Materials

Carboxydextran-coated SPIONs (ferucarbotran, SHU 555A) have previously been extensively used by us and thoroughly characterized [23,24,26,27]. Fluorescently-labelled SPIONs were purchased from MicroMod (Germany). Nanoparticles were characterized utilizing a Zetasizer Nano (Malvern Instruments). The medium particle size, polydispersity index (PDI), and zeta potential were determined. To prevent aggregation/agglomeration nanoparticles were dispersed in PBS, pH 7.4 and sonicated before each experiment.

The following fluorescent probes were used: Cellular ROS/Superoxide Detection Assay Kit (Abcam, Cambridge, UK) to detect the generation of ROS and superoxide; acridine orange (5 µg mL−1) to monitor lysosomal integrity (Thermo Fisher Scientific, Waltham, MA, USA); JC-1 (2 µM) to monitor mitochondrial membrane potential (Thermo Fisher Scientific); and ApoStat intracellular caspase detection kit (R&D Systems) to detect caspase-3 activation. To visualize the plasma membrane in confocal imaging, CellMask™ Green (Thermo Fisher Scientific) plasma membrane stain was used. The cell-permeant SYTO 13 green fluorescent nucleic acid stain (5 μM; Thermo Fisher Scientific) was used to label nuclei. Lysosomes were labeled with lysosomal marker LysoTracker® Green DND-26 (Thermo Fisher Scientific). Apoptosis was assessed using Apoptosis Detection Kit (Thermo Fisher Scientific). Nuclei were counterstained with Hoechst 33342 (Thermo Fisher Scientific). The optimal incubation time for each probe was determined experimentally. Staurosporine (STS, 2 μM) was used as a known inducer of apoptosis (Abcam).

### 4.2. Cell Culture

In this study the following cell lines were used: human hepatocellular carcinoma Huh7 (Japanese Collection of Research Bioresources, JCRB); Alexander (PLC/PRF/5, American Type Culture Collection, ATCC); and human hepatoblastoma HepG2 (ATCC). Cells were cultured in EMEM medium (ATCC) supplemented with 10% fetal bovine serum (FBS, Thermo Fisher Scientific) as recommended by the supplier. Cultures were kept in a humidified 5% CO_2_ atmosphere at 37 °C and the medium was changed once a week.

### 4.3. Pulsed Magnetic Field (PMF) System Description and PMF Treatment

In order to generate high intensity short magnetic field pulses, we utilized a previously characterized setup designed to produce magnetic fields as high as 10 T [22]. The magnetic pulse generator scheme is shown in Appendix A. The PMF generator consists of a pulsed current source based on an RCL circuit coupled with a Cu coil of inner (outer) diameter 3 (7) mm. The source of current is very compact (~10 × 10 × 10 cm^3^). The individual current pulses last for about ~15 µs. Therefore, there is no need to actively cool the coil. The maximum intensity of the magnetic field pulse produced by the PMF generator is directly proportional to the capacitor charge voltage and can reach up to 10 T at the surface of the coil [22]. For the experiments reported here, the voltage was set at 550 or 850 V so as to produce a field, respectively, of 5.5 or 8.5 T at the surface of the coil. Each pulse creates mechanical stress in the coil, and extended operation at the maximum voltage of 1000 V would shorten the life-time of the coil. It is worth noting that there is no rise in temperature at the position of the cells during the field pulse, thanks to the poor thermal contact between the cell sample holder and the coil and the relatively long dwell time between field pulses.

The drop off in magnetic field gradient along the coil axis is shown in Appendix A. The rapid drop off in both field and field gradient with distance from the coil means that cells must be positioned close to the coil to benefit from the intense magnetic field and field gradient produced. For more details of the magnetic field source, see www.pumag.fr [22].

Attached cells (Huh7, HepG2, or Alexander) seeded overnight at a density of 8000 cells per well (96-well plates) were incubated with cell culture media (EMEM 10% FBS) containing different concentrations of SPIONs (10, 50, 100 μg Fe mL^−1^) for 1.5 h at 37 °C and 5% CO_2_. Unbound SPIONs were removed by 3 consecutive washes with 200 µL of PBS once per culture well, and then fresh culture media without nanoparticles was added. After cells with incorporated nanoparticles were exposed to PMF (10 pulses of either ~8 T or ~5 T with interval 10 s); that was followed by analysis of cellular functionality.

### 4.4. Cell Viability Assay

To analyze cellular viability, we utilized the robust and commonly accepted WST-1 assay (Roche Diagnostics, Basel, Switzerland). The principle of the assay is based on the cleavage of tetrazolium salt WST-1 by cellular mitochondrial dehydrogenases. Only viable cells are able to cleave tetrazolium salt WST-1. This fact allows accurate spectrophotometric quantification of the number of metabolically active cells in the culture. We performed WST-1 assays in accordance with the manufacturer’s instructions and our established treatment protocols [53,54,59,85]. Cells were seeded onto 96-well plates at a density of 8000 cells per well, loaded with SPIONs and exposed to PMF. Twenty-four hours after the treatment, WST-1 reagent was added to each dish and it was incubated for 2 h at 37 °C to form formazan. The absorbance was measured using a TECAN microplate reader SpectraFluor Plus (TECAN, Mannedorf, Switzerland) at 450 nm.

### 4.5. Detection of Intracellular Reactive Oxygen Species (ROS)

ROS and RNS levels were assessed by a previously validated method [59,60,61,62] utilizing a Cellular ROS/Superoxide Detection Assay Kit (Abcam, Cambridge, United Kingdom). Following PMF treatment, cells were stained with a fluorescent probe labelling total ROS (green fluorescence) and a distinct probe specific to superoxide anion (orange fluorescence), according to the manufacturer’s instructions (Abcam, Cambridge, United Kingdom). A Bio Rad MRC-1024 confocal system was used to measure fluorescence intensity in labelled cells.

### 4.6. Caspase-3 Activity Assay

An ApoStat intracellular caspase detection kit (R&D Systems) was used to assess caspase-3 activation. Prior to PMF treatment, cells at a density of 8000 cells per well were seeded onto 96-well black/clear bottom plates (BD Biosciences), loaded with SPIONs and exposed to PMF. Post PMF treatment, cells were labeled with ApoStat intracellular caspase fluorescent probe and analyzed in accordance with the manufacturer’s guidelines. A TECAN microplate reader SpectraFluor Plus was used to measure the fluorescence.

### 4.7. Detection of Apoptosis

Signs of early apoptosis, namely, phosphatidylserine expression and membrane permeability, were assessed utilizing the Dead Cell Apoptosis Kit (Thermo Fisher Scientific) as described previously [53,54,59,62,85]. PMF-treated cells were labelled with Dead Cell Apoptosis Kit following the manufacturer’s guidelines. The Dead Cell Apoptosis Kit consists of Alexa Fluor 488 Annexin V, which detects phosphatidylserine expression, and propidium iodide, which is used to assess membrane permeability enabling discretion of necrotic cells. Nuclei were labeled with Hoechst 33342. After staining, cells were fixed by 4% paraformaldehyde. Treatment with 2 µM staurosporine for 4 h was used as a positive control. An epifluorescent microscope IM-2FL (Optika Microscopes, Ponteranica, Italy) was utilized to acquire fluorescence images. ImageJ software was used for image processing and fluorescent micrograph quantification.

### 4.8. Quantification of Mitochondrial Membrane Potential

We used the JC-1-based previously described methodology to assess changes in the mitochondrial membrane potential (ΔmΦ) [52,53,54]. Cells were seeded onto 96-well clear bottom plates (BD Biosciences) at a density of 8000 cells per well. After cells were incubated with cell culture media (EMEM 10% FBS) containing different concentrations of SPIONs (10, 50, 100 μg Fe mL^−1^) for 1.5 h at 37 °C and 5% CO_2_. Cells with incorporated nanoparticles were stained with 2 µM JC-1 probe and then exposed to PMF (10 pulses of either ~8 T or ~5 T at an interval of 10 s). Assessment of the mitochondrial potential was done utilizing a Bio-Rad MRC-1024 confocal imaging system. JC-1 is more selective in labeling mitochondria in comparison with other cationic dyes [86,87]. The dye is mitochondria permeable where it accumulates and reversibly changes color from red to green as the membrane potential decreases. In healthy cells with intact mitochondria and high mitochondrial ΔmΦ, JC-1 associates with J-aggregates, to produce intense red fluorescence On the contrary, in apoptotic or unhealthy cells with damaged mitochondria and low ΔmΦ, JC-1 remains in the monomeric form, which shows only green fluorescence. Thus, the ratio of green to red fluorescence can be used as an indicator of mitochondria damage and ΔmΦ. Importantly, this ratio is not influenced by other factors, such as mitochondrial size, shape, and density, which may significantly affect fluorescence signals derived from single-component probes [86,87]. In order to quantify ΔmΦ, cells were assessed by laser scanning confocal microscopy and subsequently analyzed utilizing LaserSharp 2000 software (Bio-Rad, Cambridge, UK), as described previously [52,53,62].

### 4.9. Assessment of Lysosomal Integrity by Acridine Orange (AO)

Cells were seeded onto 96-well clear bottom plates (BD Biosciences) at a density of 8000 cells per well. After, cells were incubated with cell culture media (EMEM 10% FBS) containing different concentrations of SPIONs (10, 50, 100 μg Fe mL^−1^) for 1.5 h at 37 °C and 5% CO_2_. Cells with incorporated nanoparticles were labeled with 5 µg mL^−1^ AO in DMEM culture medium for 15 min at 37 °C and then exposed to PMF (10 pulses of either ~8 T or ~5 T at interval of 10 s). Following PMF treatment, cells were cultured at 37 °C for indicated periods of time and the intensity of orange fluorescence was then measured using a TECAN microplate reader SpectraFluor Plus. Readings were done in quadruplicates.

### 4.10. Immunofluorescence Staining

Cells were seeded on μ-Slides (Ibidi, Martinsried); then, incubated with cell culture media (EMEM 10% FBS) containing different concentrations of SPIONs (50 μg Fe mL^−1^) for 1.5 h at 37 °C and 5% CO_2_. Cells with incorporated nanoparticles were exposed to PMF (10 pulses of either ~8 T or ~5 T at intervals of 10 s). After 4 h, cells were fixed in 4% paraformaldehyde in PBS for 15 min and stained for cathepsin B (1:400, Cell Signaling, catalogue no. 31718) and LAMP1 (1:100, Cell Signaling, catalogue no. 15665). Fixed cells were imaged using a Bio-Rad MRC-1024 confocal system (Bio-Rad, Cambridge, MA).

### 4.11. Generation of 3D Multicellular Aggregates

3D multicellular spheroids were prepared by the hanging drop technique [88,89]. Briefly, Huh7, HepG2 and Alexander cells were re-suspended in complete culture medium (EMEM, containing 10% FBS) at a density of 10^6^ cells per mL and the 35 µL drops, containing 3.5 × 10^4^ cells, were distributed on the lid of a 100 mm Petri dish. Then, the lid was inverted over the PBS-filled bottom part of the dish and incubated at 37 °C with 5% CO_2_ and 95% humidity for 72 hrs. After 3 days of culture, the individual multicellular aggregates from each drop were collected and used for further studies. 3D multicellular aggregates were incubated with cell culture media (EMEM 10% FBS) containing different concentrations of SPIONs (100 μg Fe mL^−1^) for 2 h at 37 °C and 5% CO_2_. 3D multicellular aggregates with incorporated nanoparticles were exposed to PMF (10 pulses of ~8 T with interval 10 s). After PMF treatment apoptosis analysis was assessed using the Dead Cell Apoptosis Kit (Thermo Fisher Scientific) as described above.

### 4.12. Statistical Analysis

Quantitative results were assessed with ANOVA statistics and presented in the form of mean ± SEM. ANOVA Fisher’s LSD and Newman-Keuls tests were utilized to decipher the statistical significance between the compared groups. Differences were considered statistically significant at ** p* < 0.05.

We used defined and established guidelines for quantitative fluorescence microscopy analysis (analysis of lysosomal size, colocalization) [90]. In order to determine sample size, we utilized a previously described statistical method [91]. According to this method, the sample size for 95% confidence interval and 0.9 statistical power corresponds to 30. Thus, 30 cells were used in fluorescence microscopy quantification.

## 5. Conclusions

In summary, using a benchtop PMF system, we can exert large enough magnetic gradient force on clusters of SPIONs to induce leakage of lysosomal cathepsins B into the cytoplasm. We estimated a force of 500 pN as a minimal effective force to induce LMP. Initiated by PMF, lysosomal leakage results in mitochondrial damage, which in turn leads to apoptosis execution and subsequent cell death. Furthermore, we validated our approach on a three-dimensional culture model mimicking in vivo conditions. We are well aware of the limitations of our system. The rapid decay of the field gradient produced by the pulsed field source with distance from the coil precludes full usage of the current system in real in vivo applications. However, the pulsed field system generates a magnetic field that theoretically should be able to penetrate deep tissue. Furthermore, the resultant magnetic field gradient is expected to produce a large enough magnetic gradient force. Of course, full adaptation of our PMF system for real in vivo testing requires further thorough investigations and tuning. However, we would like to emphasize that previous studies utilizing alternating/dynamic magnetic fields to disrupt membranes of SPION-loaded cancer cells showed neither in vivo nor three-dimensional culture validation of magnetic field-induced cell death.

Our study clearly showed that PMF treatment of SPION-loaded cells represents a fast, feasible and noninvasive tool to remotely control apoptosis execution in distinct liver cancer cells. The presented PMF system can serve as a platform to develop new technologies for a wide range of distinct biomedical applications.

## Figures and Tables

**Figure 1 cancers-11-01873-f001:**
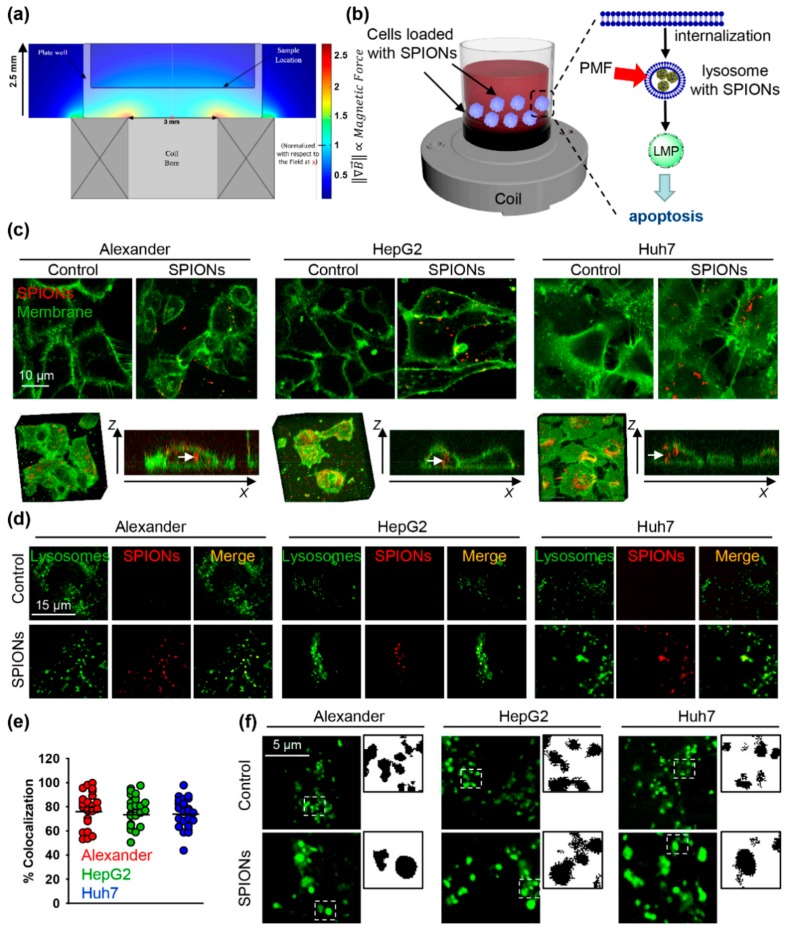
(**a**) Simulation of the spatial distribution of the norm of the field gradient produced by the pulsed field system, which is proportional to the force exerted by the system on a target object. The values are normalized with respect to the value at the center of the well base. The external dimensions of a well from a 96-well plate are overlaid. (**b**) The concept of lysosomal membrane permeabilization by superparamagnetic iron oxide nanoparticles (SPIONs) in a pulsed magnetic field. Nanoparticles are taken up into endosomes and lysosomes due to receptor mediated endocytosis. When the pulsed magnetic field (PMF) is activated at this point, the clusters of nanoparticles exert mechanical forces that cause injury to the lysosomal membrane. This in turn causes leakage of the lysosomal contents into the cytoplasm, leading to a decrease in its pH and subsequent apoptosis. (**c**) Uptake of fluorescently labeled SPIONs by Huh7, HepG2, and Alexander cells. Cells were treated for 1.5 h with SPIONs 50 μg Fe mL^−1^ (red). Cell membranes were labeled with CellMask™ Green (green). Labeled cells were then imaged by confocal microscopy, and the image was processed using ImageJ software (NIH). 3D reconstruction and orthogonal projections of SPION-labelled cells were done using ImageJ software. (**d**) Localization of fluorescently labeled SPIONs in lysosomes. Cells were treated as in (**c**). Lysosomes were labeled with lysosomal marker LysoTracker^®^ Green DND-26 (green). Labeled cells were then imaged by confocal microscopy, and the image was processed using ImageJ software. Nanoparticles are red, colocalization is yellow. Colocalization analysis of SPIONs and lysosomes from images (**d**) is presented in (**e**). Quantifications performed using ImageJ are presented as means of *n* = 34 cells. (**f**) High resolution confocal microscopy of cells loaded with SPIONs and labelled with lysosomal marker LysoTracker® Green DND-26 (green). Binarization of the selected region was done using ImageJ software.

**Figure 2 cancers-11-01873-f002:**
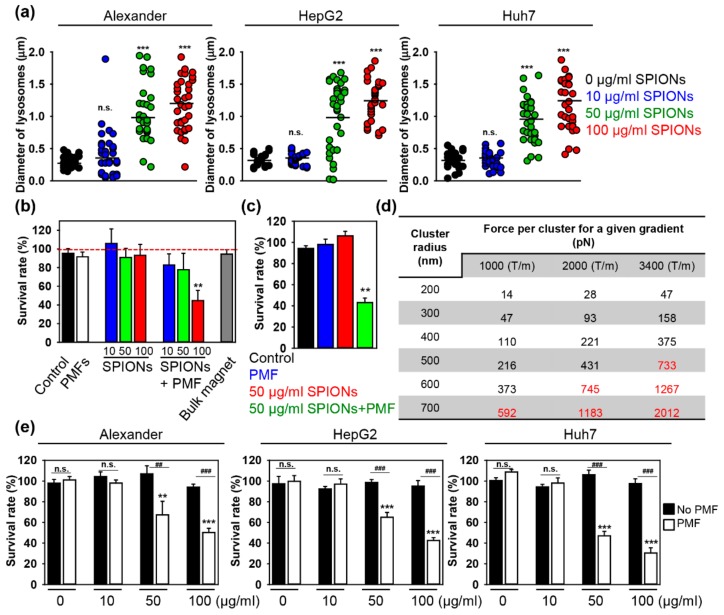
(**a**) Assessment of the lysosomal size upon SPION uptake. Huh7, HepG2, and Alexander cells were treated for 1.5 h with indicated concentrations of SPIONs. Lysosomes were labeled with lysosomal marker LysoTracker® Green DND-26 (green). Labeled cells were then imaged by confocal microscopy, and images were quantified using ImageJ software. Quantifications performed using ImageJ are presented as means of *n* = 34 cells. *** *p* < 0.001 denote significant differences with respect to control (no particles, no PMF treatment). (**b**) Huh7 cells were pre-incubated with different concentrations of SPIONs (10, 50, 100 μg Fe mL^−1^) for 1.5 h. After that cells with incorporated nanoparticles were exposed to PMF (10 pulses of ~5 T at intervals of 10 s). the 24 h cell viability was assessed by the WST-1 assay. The data were normalized to control values (no particles, no PMF exposure) and expressed as means ± SDs, *n* = 3 each. ** *p* < 0.01 denote significant differences with respect to control (no particles, no PMF treatment). (**c**) Huh7 cells were pre-incubated with SPIONs (50 μg Fe mL^−1^) for 1.5 h. After that, cells with incorporated nanoparticles were exposed to PMF (10 pulses of ~8 T at intervals of 10 s). The 24 h cell viability was assessed by the WST-1 assay. The data were normalized to control values (no particles, no PMF exposure) and expressed as means ± SEMs, *n* = 3 each. ** *p* < 0.01 denote significant differences respect to control (no particles, no PMF treatment). (**d**) Estimations of the magnetic gradient force exerted on clusters of SPIONs. (**e**) Huh7, HepG2, and Alexander cells were pre-incubated with different concentrations of SPIONs (10, 50, 100 μg Fe mL^−1^) for 1.5 h. After, cells with incorporated nanoparticles were exposed to PMF (10 pulses of ~ 8 T at intervals of 10 s). The 24 h cell viability was assessed by the WST-1 assay. The data were normalized to control values (no particles, no PMF exposure) and expressed as means ± SDs, *n* = 3 each. ** *p* < 0.01 and *** *p* < 0.001 denote significant differences with respect to control (no particles, no PMF treatment).

**Figure 3 cancers-11-01873-f003:**
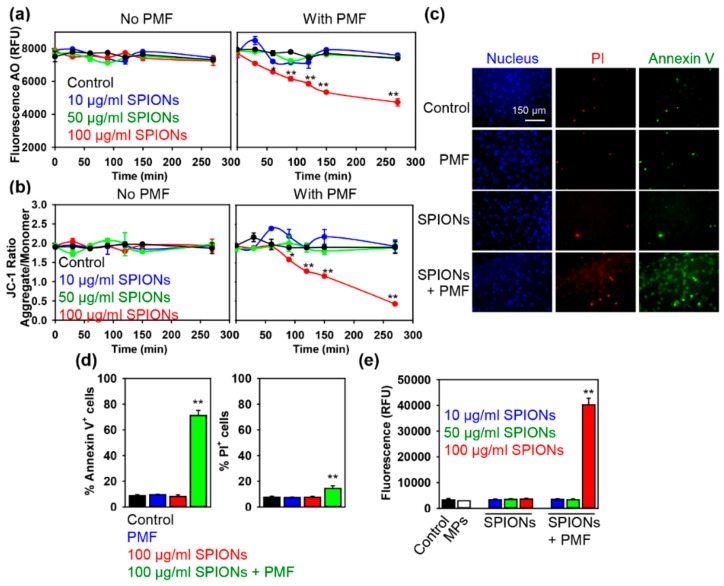
(**a**) Huh7 cells were pre-incubated with different concentrations of SPIONs (10, 50, 100 μg Fe mL^−1^) for 1.5 h. After cells with incorporated nanoparticles were exposed to PMF (10 pulses of ~5 T at intervals of 10 s) and stained with acridine orange (AO), the fluorescence intensity was measured using a fluorescent microplate reader. The data are expressed as means ± SEMs, *n* = 3 each. * *p* < 0.05 and ** *p* < 0.01 denote significant differences with respect to controls (no particles, no PMF treatment). (**b**) Huh7 cells were treated as in (**a**). After PMF treatment cells were stained with 2 µM JC-1 for 30 min and analyzed by fluorescent microplate reader. The data are expressed as means ± SEMs, *n* = 3 each. * *p* < 0.05 and ** *p* < 0.01 denote significant differences with respect to controls (no particles, no PMF treatment). (**c**) Huh7 cells were pre-incubated with SPIONs 100 μg Fe mL^−1^ for 1.5 h. After that cells with incorporated nanoparticles were exposed to PMF (10 pulses of ~5 T at intervals of 10 s); then, 4 h after treatment cells were labelled with Hoechst nuclear stain—blue dye, annexin V—green dye, and propidium iodide (PI)—red dye. Labelled cells were imaged with fluorescence microscopy. Representative images out of three independent experiments are shown. (**d**) Apoptosis assessment in PMF-treated Huh7 cells. Cells were treated as in (**c**); then, 4 h after the treatment cells were labelled with Hoechst nuclear stain, annexin V, and propidium iodide. Fluorescence microscopy was used to visualize fluorescently stained cells. The percentage of Annexin V and PI positive cells was calculated with ImageJ (NIH). The data are expressed as means ± SEMs, *n* = 3 each. ** *p* < 0.01 denote significant differences with respect to control (no particles, no PMF treatment). (**e**) Huh7 cells were pre-incubated with different concentrations of SPIONs (10, 50, 100 μg Fe mL^−1^) for 1.5 h. After that, cells with incorporated nanoparticles were exposed to PMF (10 pulses of ~5 T at intervals of 10 s), and then caspase-3 activity was assessed using an ApoStat detection kit (R&D Systems) and analyzed with a fluorescent microplate reader. The data are expressed as means ± SEMs, *n* = 3 each. ** *p* < 0.01 denote significant differences with respect to control (no particles, no PMF treatment).

**Figure 4 cancers-11-01873-f004:**
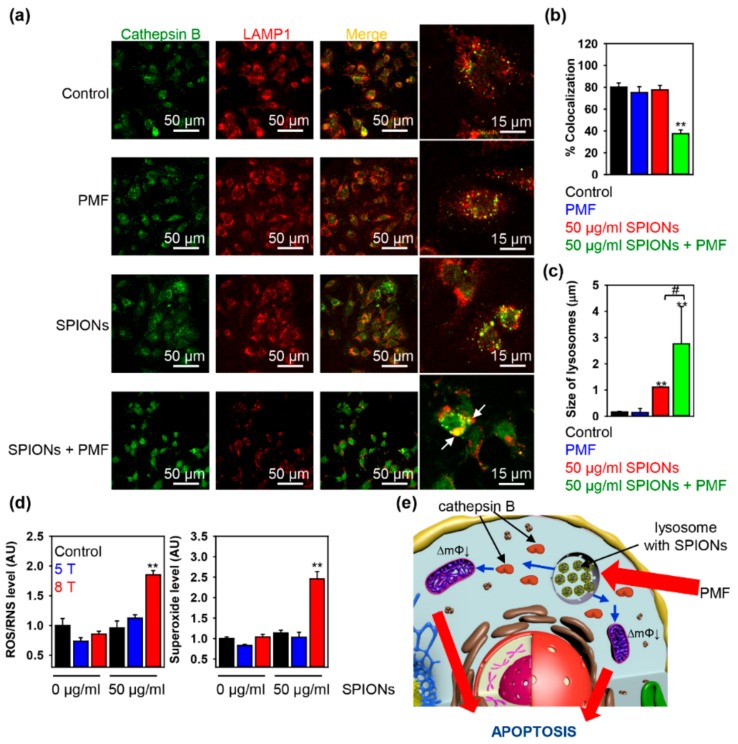
(**a**) Huh7 cells were pre-incubated with SPIONs 50 μg Fe mL^−1^ for 1.5 h. After incubation, cells with incorporated nanoparticles were exposed to PMF (10 pulses of ~8 T at intervals of 10 s), and then 4 h after treatment, cells were analyzed by confocal microscopy using LAMP1 antibody as a marker of lysosomes (red) and cathepsin B (green). Colocalization is shown in yellow. Colocalization analysis of cathepsin B and LAMP1 from images (**a**) is presented in (**b**). ** *p* < 0.01 denote significant differences with respect to control (no particles, no PMF treatment). (**c**) Assessment of the lysosomal size upon PMF treatment. Cells were treated as in (a). Images were quantified using ImageJ software. ** *p* < 0.01, # *p* < 0.05. (**d**) Intracellular ROS/superoxide (O_2_^−^) production upon PMF treatment. Huh7 cells were pre-incubated with SPIONs 50 μg Fe mL^−1^ for 1.5 h. After, cells with incorporated nanoparticles were exposed to PMF (10 pulses of either ~8 T or ~5 T at intervals of 10 s), followed by ROS measuring, using the cellular ROS/RNS detection kit (Abcam) by spectrofluorometry. ** *p* < 0.01 denote significant differences with respect to control (no particles, no PMF treatment). (**e**) Principle of remote induction of apoptosis by PMF. Schematic representation of lysosomal membrane permeabilization by SPIONs in a pulsed magnetic field.

**Figure 5 cancers-11-01873-f005:**
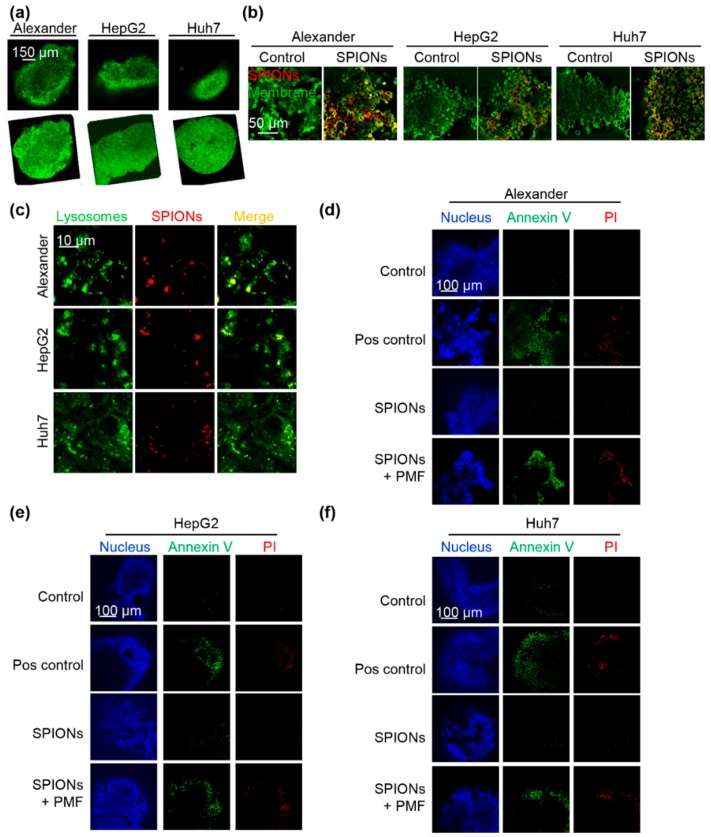
(**a**) Organoid-like 3D structures from Huh7, HepG2, and Alexander cells. Nuclei were labeled with SYTO 13 green fluorescent nucleic acid stain (green). Labeled cells were then imaged by confocal microscopy, and 3D images were constructed using ImageJ software. (**b**) Organoid-like 3D structures form Huh7, HepG2, and Alexander cells were treated for 2 h with SPIONs 100 μg Fe mL^−1^ (red). CellMask™ Green (green) was used to label cell membranes. Labeled cells were then imaged by confocal microscopy, and the image was processed using ImageJ software (NIH). (**c**) Organoid-like 3D structures form Huh7, HepG2 and Alexander cells were treated for 2 h with SPIONs 100 μg Fe mL^−1^ (red). LysoTracker® Green DND-26 (green) was used to label lysosomes. Labeled cells were then imaged by confocal microscopy, and the image was processed using ImageJ software. Organoid-like 3D structures from Alexander (**d**), HepG2 (**e**), and Huh7 (**f**) cells were treated for 2 h with SPIONs 100 μg Fe mL^−1^. After cells with incorporated nanoparticles were exposed to PMF (10 pulses of ~8 T at intervals of 10 s), then 6 h after treatment cells were labelled with Hoechst nuclear stain—blue dye, annexin V—green dye, and propidium iodide—red dye. Labelled cells were imaged by confocal microscopy. Representative images out of three independent experiments are shown. Positive control—2 µM staurosporine for 6 h.

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
