# Peer review of "Remote Actuation of Apoptosis in Liver Cancer Cells via Magneto-Mechanical Modulation of Iron Oxide Nanoparticles"

_cancers, 2019, doi:10.3390/cancers11121873_

Round 1
Reviewer 1 Report
In this study, Lunov, et al. demonstrated the potential of pulsed magnetic fields with high field gradients to induce cell death in the in vivo relevant models of liver cancers via mechanical actuation of magnetic nanomaterials and subsequent disruption of lysosomal membrane. The manuscript is well written and the results are well discussed and explained in detail. I, therefore, recommend publication with minor revision. I would ask the authors to consider the following minor comment before acceptance.
How about the selectivity of this approach towards healthy cells and cancer cells? Since these nanoparticles are not actively targeted, how can they selectively kill the cancer cells?
Line 405: typo; "first ttempt"
Author Response
POINT BY POINT REPLY TO THE REVIEWER’S COMMENTS
We would like to thank the Reviewer for his/her careful and rigorous review and constructive criticism that helped us to improve the quality of our paper.
Detailed point-by-point responses to all Reviewer’s remarks together with the corresponding amendments made to the manuscript are provided below.
Reviewer’s questions and comments are given in italic; replies are given in blue and the changes in the manuscript are marked in red.
Reviewer #1 (Comments to the author):
Summary:
In this study, Lunov, et al. demonstrated the potential of pulsed magnetic fields with high field gradients to induce cell death in the in vivo relevant models of liver cancers via mechanical actuation of magnetic nanomaterials and subsequent disruption of lysosomal membrane. The manuscript is well written and the results are well discussed and explained in detail. I, therefore, recommend publication with minor revision. I would ask the authors to consider the following minor comment before acceptance.
Minor comments:
How about the selectivity of this approach towards healthy cells and cancer cells? Since these nanoparticles are not actively targeted, how can they selectively kill the cancer cells?
We are thankful the Reviewer for that comment. Indeed, targeting and selectivity of nano-based drugs represents a challenge. However, EPR effect can be effectively used in order to passively target tumors. We elaborated our response in the Discussion part. Page#12:
It is worth mentioning here that for potential in vivo translation of such cancer treatment, one needs to consider targeting strategies and the selectivity of the treatment. Targeting is crucial in light of recent criticism of different nanoparticle targeting approaches [31]. However, enhanced permeability and retention (EPR) may help to overcome targeting challenges [66]. The EPR effect is based on the preferential passive accumulation of nanoparticles in cancerous tissues due to the enhanced permeability of the vasculature that supplies such tissues [66-68]. In fact, liposomal doxorubicin (Doxil™/Caelyx™), that utilizes the EPR effect, was approved by the FDA and is used in clinics as a treatment modality [66,69]. The EPR effect may possess a significant advantage for liver cancer targeting, since dextran-decorated SPIONs are known to be passively accumulated in liver, for a review see [70]. Additionally, cancer treatment strategies involving LMP have selectivity advantages. Recent studies show that lysosomal membranes of cancer cells are mechanically weaker than those of noncancerous cells [71,72]. Thus, cancer cells are more susceptible to LMP. In fact, a number of studies have already utilized this approach to selectively sensitize different cancer cell types to cell death [47,73-76]. Indeed, several agents that target LMP in cancer are being extensively investigated in preclinical studies [71].
Line 405: typo; "first ttempt"
We changed accordingly.
Reviewer 2 Report
P1. Abstract: “three-dimensional culture model”. I would add “cellular” adjective here.
P4. First para. Which parameter is more important, the absolute value of the magnetic field B or the gradient dB/dz? The equation on P6 clearly states that the gradient is more relevant. Accordingly please explain what is the impact of the magnetic field induction? Why does it matter?
P6. First para. The authors estimate the maximum magnetic field gradient to 3400, 2000 and 1000 T/m. The authors should clarify the description in Fig.S1b where the maximum field gradient was shown match dB/dz~700 T/m.
P6. Second para. “A recent study estimated a force of 500 pN”. Is it possible to compare it with the viscosity Stokes drag force given the medium viscosity, temperature and hydrodynamic radius of the SPION particle (~500 nm)? If so, what would be the translation distance upon 10 s field gradient pulse with respect to the lysosome size (Fig.2a). How would the authors relate the pressure (roughly
General remarks:
The article is well written. The language is clear. The figure quality is bad.
Author Response
POINT BY POINT REPLY TO THE REVIEWER’S COMMENTS
We would like to thank the Reviewer for his/her careful and rigorous review and constructive criticism that helped us to improve the quality of our paper.
Detailed point-by-point responses to all Reviewer’s remarks together with the corresponding amendments made to the manuscript are provided below.
Reviewer’s questions and comments are given in italic; replies are given in blue and the changes in the manuscript are marked in red.
Reviewer #2 (Comments to the author):
Minor comments:
P1. Abstract: “three-dimensional culture model”. I would add “cellular” adjective here.
We changed the text accordingly.
P4. First para. Which parameter is more important, the absolute value of the magnetic field B or the gradient dB/dz? The equation on P6 clearly states that the gradient is more relevant. Accordingly please explain what is the impact of the magnetic field induction? Why does it matter?
To cope with reviewer’s comments, we added brief discussion with reference on the relevant issue in the text accordingly. Page#6:
Note that even higher field gradients can be achieved with low and/or moderate magnetic field intensities if the size of the field source is significantly reduced [38], but the range of interaction would also be reduced.
P6. First para. The authors estimate the maximum magnetic field gradient to 3400, 2000 and 1000 T/m. The authors should clarify the description in Fig.S1b where the maximum field gradient was shown match dB/dz~700 T/m.
We added a section elaborating the raised question in the revised version of the manuscript accordingly with the Reviewer’s remarks as suggested. See Page#6, please.
Figure S1b depicts the drop-off in field gradient with distance from the surface of the coil for a charge of 100 V. Importantly, at the distance of ~ 1 mm, where cells are located, the gradient is reduced by approximately by 40%. Given that the field intensity and resultant gradient are proportional to the voltage, one can roughly estimate the field gradient at the position of the cells to be ~ 3400, 2000 and 1000 T/m, for voltages of 850, 550 and 300 V, respectively.
P6. Second para. “A recent study estimated a force of 500 pN”. Is it possible to compare it with the viscosity Stokes drag force given the medium viscosity, temperature and hydrodynamic radius of the SPION particle (~500 nm)? If so, what would be the translation distance upon 10 s field gradient pulse with respect to the lysosome size (Fig.2a). How would the authors relate the pressure (roughly
We are thankful the Reviewer for constructive criticism, which have helped us to improve the data and presentation of the manuscript. We agree with the Reviewer that pressure and Stokes’ drag force estimations are important. Thus, we added relevant calculations and new references in the revised version of the manuscript. See Page#6, please.
Additionally, one can compare this force with the Stokes’ drag force, Fs = 6πµRv, where Fs is known as Stokes' drag force, µ represents the viscosity, R is the radius of the vesicle and v is the flow velocity. Assuming the lysosome is spherical with a radius of 500 nm, and that it is moving through the cytosol of density 1.3∙10-3 Pa∙s [40] at a velocity of 10 µm/s, one can roughly estimate a drag force of ~ 0.1 pN. It is obvious that the force exerted on magnetic nanoparticles by the applied magnetic field (500 pN) outweighs the Stokes’ drag force. However, one needs to take into account that lysosomes are not freely moving in the cytosol [41]. Indeed, lysosomes are linked to the cytoskeletal elements [41,42]. Therefore, it is important to assess the hoop stress (σ) on a thin-walled lysosomal vesicle. One can roughly derive the hoop stress as σ = P∙r/2∙t, where P is the internal pressure (knowing the exerted force ~ 500 pN and lysosomal diameter ~ 1 μm, one can estimate the pressure to be ~ 600 Pa), r represents the radius of the lysosome, t is the lysosomal membrane thickness. Assuming the lysosomal radius ~ 500 nm (Figure 2a) and membrane thickness of ~ 10 nm [43], we roughly estimated the hoop stress to be about 15 kPa. This estimation is perfectly in line with the recently measured 12 kPa critical pressure required for membrane rupture in endothelial cells [44].
The article is well written. The language is clear. The figure quality is bad.
We improved quality of the figures as requested.